# How Does Digital Transformation Impact Green Supply Chain Development? An Empirical Analysis Based on the TOE Theoretical Framework

**Weimin Li, Xiaoyu Xiao, Xinyue Yang * and Li Li**

School of Economics, Liaoning University, Shenyang 110036, China; liweimin@lnu.edu.cn (W.L.);
xiaoxy97118@gmail.com (X.X.); jake15140189696@126.com (L.L.)
* Correspondence: yangxinyue97@163.com

**Abstract:** Digital transformation and sustainability are both at the forefront of current supply chain developments. However, the specific mechanisms of how digital transformation and green supply chain development interact still need to be clarified, which can help supply chain business operators to enhance supply chain sustainability more effectively. This paper focuses on how the companies' organization structure and the socio-economic environment interact with digital technologies under the process of green supply chain development. Based on the "Technology–Organization–Environment" (TOE) framework, this paper analyze how digital transformation can drive green supply chain development. To test the TOE theoretical analysis framework, this paper calculates the digital transformation and green supply chain development index at the provincial level in China and conducts an empirical study. The main findings and implications of this paper can be summarized in the following aspects: First, according to the TOE theory, the external environment dimensions, such as the market and policy environments, affect the role of digital technology in promoting GSC development. Second, in the organizational dimensions, labor–capital relations, company size, and ownership factors can all affect the contribution of digital transformation to green supply chains. Third, there are differences in the impact of different types of digitization technologies on GSC development.

**Keywords:** digital transformation; green supply chain; the TOE theory; China



## 1. Introduction

The impact of digital technologies on industrial sustainability is an emerging topic of discussion [1,2]. Digital transformation and sustainability are both at the forefront of current supply chain developments. However, how digital technologies contribute to sustainability of supply chains in specific industries needs to be explored more. Transportation is one of the world's most carbon-emitting industrial sectors. According to the IEA, approximately 24% of the world's carbon emissions came from the transportation sector in 2021. Achieving sustainable development has become an important direction for the current development of the transportation industry worldwide. In this context, the concept of a green supply chain was proposed, which refers to the reduction of carbon emissions in the transportation chain through advanced technology and effective management. Recently, studies have focused on the application of digital technologies in green supply chains [3–5]. However, related studies are mainly qualitative or case studies, and there is a relative lack of quantitative studies.

At the same time, as an act of corporate innovation, digital transformation is not only linked to the level of technology, but also to the social and economic environment. In the "Technology–Organization–Environment" (TOE) theoretical framework on the factors influencing business decisions, the market environment is considered to be an important dimension of the external environment. Studies have been conducted to examine the

influencing factors of digital transformation in supply chains in the form of case studies under the TOE framework [6–8]. However, the specifics of the impact of the market environment are vague in the case studies under the TOE framework. One of the existing research gaps is that more quantitative evidence is needed on the impact of the market environment on firm digital transformation decisions under the TOE framework.

There are two aspects of existing research gaps that need to be filled: on the one hand, although scholars have recognized the important role of digital transformation in promoting sustainable industrial development [9], there are only a few cases and evidence from transport sectors. More specifically, how do digital technologies affect the development of green supply chains? On the other hand, a number of relevant conceptual studies and case studies have been enriched since the TOE theoretical framework was proposed. However, quantitative research based on TOE is relatively scarce, especially on how technology interacts with business models in the recent digital era [10,11]. Digital transformation and green supply chain development is an organic system involving internal and external factors such as enterprise, society, and technology, which requires system theory to explain and analyze the specific interaction mechanism [9].

Therefore, this paper views the interaction between digital transformation and green supply chain development as a system influenced by three types of factors: technological, organizational, and environmental, and we adopt a theoretical and structural empirical approach based on the TOE framework to study this interactive process. This paper, therefore, introduces and extends the TOE analytical framework to specifically analyze how digital transformation can drive green supply chain development. We focus on how the companies' organization structure and the socio-economic environment interact with digital technologies under the process of green supply chain development. To test the TOE theoretical analysis framework, this paper calculates the digital transformation and green supply chain development index at the provincial level in China and conducts an empirical study. The empirical research in this paper will quantify the impact of the interaction between the organization, the environment, and digital technology during the green supply chain development.

China has both the world's largest transportation system and the largest carbon footprint of the transportation sector and is also under pressure to develop transportation sustainably [12,13]. As of 2022, the transport sector is the second largest emitter of carbon in China, and the share of transport carbon emissions has been increasing year on year for the last decade. To address these problems, the Chinese government has proposed a number of emission reduction targets such as "carbon peaking" and "carbon neutrality" and has identified transport as an important area for reducing emissions. In 2022, China's Ministry of Transport released the latest version of the Green Transport Standard System, which clearly identifies green supply chain construction as a key development strategy [14]. Therefore, the study of this paper can provide some policy implications into achieving emission reductions in the transport sector through digital technology. Additionally, this paper's research on China's digital transformation and green supply chains can also provide some insights for other countries that have the pressure to reduce transport emissions.

The present study is organized as follows. Section 2 is a literature review. In this section, we will provide a concise overview of the literature on digital technologies and green supply chains, digital transformation and sustainable development, the "technology-organization-environment" (TOE) theory, and present possible novel points for this paper. Section 3 presents the theoretical framework, where we introduce the TOE theoretical framework to analyze how the three dimensions of digital technology, organization characteristics, and socio-economic environment interact during the process of green supply chain development. Section 4 displays the data and measuring method we used to calculate the proxy variables on the level of regional digital transformation, green supply chain development, and other socio-economic environmental dimensions and organizational dimension variables. Section 5 reports the results of the empirical study of the TOE framework. Section 6 summarizes the main conclusions and research outlooks of this paper.

## 2. Literature Review

### 2.1. Digital Transformation and Sustainability

Digital technologies are recognized as one of the key tools to drive sustainable development [15]. At the same time, digital technology is also widely recognized as one of the key tools to drive sustainable development, as it can provide innovative solutions to address environmental, social, and economic challenges [16,17]. This paper provides a review of recent literature on the relationship between digital technologies and sustainable development, which finds that digital technologies have a wide potential for application in areas such as energy [18,19], cleaner production [20,21], agriculture and urban planning, and can provide innovative solutions to address the challenges facing sustainable development [22–25].

For example, digital technologies play an important role in environmental monitoring, pollution control, and resource management [26]. Blockchain, AI in optimizing energy use and controlling emissions has been widely used in developed economies such as in Europe and North America [26–28]. By analyzing big data and sensor networks, digital technologies can help monitor and predict environmental changes and provide effective conservation strategies [29,30]. Particularly in the transport sector, digital technologies are already available for the optimization and intelligent management of logistics systems, including logistics networks, supply chain stability analysis, order demand forecasting, and energy consumption optimization [25–28,31–35].

However, the application of digital technologies in environmental pollution management still faces a number of challenges and risks, such as data privacy, computational errors, and technical operability [36,37]. Therefore, in order to fully utilize the role of digital technology in environmental pollution management, appropriate policies and regulations need to be developed to ensure that the application complies with the principles of environmental protection, while safeguarding the interests of society and the environment [38].

### 2.2. Digital Transformation and Green Supply Chain

As producers and consumers become more environmentally conscious, companies are increasingly required to take environmental considerations into account in their product supply chains [39]. This trend has prompted companies to seek to establish green supply chains that balance profit maximization with environmental objectives [40]. Digital technology offers new solutions for the establishment of green supply chains. The application of digital technologies has been widely used in the field of green supply chains [34–36,41–43]. The existing literature focuses on the following areas: green logistics, lifecycle analysis, supply chain transparency and traceability, and partner management.

Firstly, green logistics. Digitization technologies such as the Internet of Things can be used to monitor information on energy consumption, emissions, and traffic congestion during the transportation of goods, and, thus, optimize the way goods are transported [44,45]. In addition, digital technologies can enable the tracking and management of goods and assets, as well as the forecasting of demand in all parts of the supply chain [46]. Secondly, life cycle analysis: life cycle analysis is an important tool in green supply chains to assess the environmental impact of products throughout their life cycle [47]. From raw material procurement to disposal, Leng et al. [48] and Brandín et al.'s [49] study found that digital technologies such as big data and cloud computing can collect and analyze this data to provide companies with a basis for sustainability decisions. Thirdly, supply chain transparency and traceability: supply chain transparency and traceability are critical to the establishment of green supply chains [50]. Blockchain can ensure traceability and transparency at every point in the supply chain [51,52]. Fourthly, partner management is an essential external environment in a green supply chain [53,54]. Digital technologies such as supply chain networks can improve collaboration and communication between companies and their partners, thus optimizing processes and efficiency in the supply chain [55].

This paper shows that digital technologies can not only improve environmental issues in green supply chains, but also increase the efficiency and flexibility of supply chains. The

use of digital technologies in green supply chains can improve transparency, traceability, and efficiency for companies, thereby balancing best interests with environmental objectives. However, a number of challenges remain in this area, such as high energy consumption of digital equipment, information sharing, and data security issues.

*2.3. The "Technology-Organization-Environment" (TOE) Theory*

The TOE framework proposes that technological, organizational, and environmental factors are critical in explaining technology adoption at an organizational level [56,57]. Technological factors include technical standards and enterprise capabilities, while organizational factors include corporate strategy, organizational structure, ownership type, manager awareness, and enterprise resources. Environmental factors include institutional environment, economic, social, and cultural aspects [58]. These factors have an impact on the operation and development of a business at different stages and areas of business management.

Firstly, the interaction between technology and organization is central in business management [56–58]. Technological innovation is one of the keys to a company's success, while the organization is the key to realizing technological innovation and translating it into business value. For example, when a company wants to adopt a new technology, it must assess whether the technology meets the company's goals and needs and decide how to adapt it. In addition, the organizational structure and culture within the company must be adapted accordingly to ensure that the technology is used effectively. The interaction between technology and organization is, therefore, crucial in business management.

Secondly, the socio-economic environment is also an important factor in business management. The socio-economic environment includes a number of aspects such as laws and regulations, political stability, culture and social values. These factors can affect a company's market opportunities and competitive environment in a particular country or region. For example, government policies and regulations can encourage or restrict the growth and innovation of a business [59], and cultural and social values may also influence consumer purchasing decisions and market demand. Therefore, companies need to take into account the impact of the socio-economic environment when making management decisions.

Finally, the interaction between technology, the organization, and the socio-economic environment is also important. For example, in the field of emerging technologies, the technology itself often has far-reaching effects on the organization and society. These impacts may include redesigning organizational structures, changing employee roles and competency needs, and adapting a company's business model. At the same time, organizational structures and cultures can facilitate or hinder the adoption and development of new technologies. The socio-economic environment can also influence the opportunities and challenges for companies in the field of emerging technologies, such as policy incentives and financial support.

In practice, business managers need to make decisions by striking a sensible balance between the technological, organizational, and socio-economic environments. They need to understand how these factors interact in order to make effective business plans and decisions. By appropriately applying knowledge of the three dimensions of technology, organization, and socio-economic environment, businesses can better adapt to market changes and succeed [60].

Recently, the TOE framework emphasized the importance of leaders, organizational structures, and external environments in the diffusion of innovative technologies [61]. The adoption of new technologies has been an important topic of research in the field of organizational studies. The Technology–Organization–Environment (TOE) framework and the Diffusion of Innovation (DOI) framework have been widely used to explain technology adoption phenomena [62]. There are also studies that use the TOE framework to analyze issues in supply chain management and include digital technologies (e.g., blockchain, AI) as an important element of the technology dimension [27,28,63,64]. While existing studies

have explored technology adoption and its influencing factors, the unique features of digital transformation in the green supply chain require further investigation. This paper aims to explore the driving factors of digital transformation adoption in the green supply chain development under the TOE theoretical framework.

### 3. Theoretical Framework

According to the TOE theory, digital technology has become a key driver of green supply chain development, not only through external environment factors such as the market and policies but also in terms of business organization. The theoretical mechanisms of these interactions can be summarized as in Figure 1.

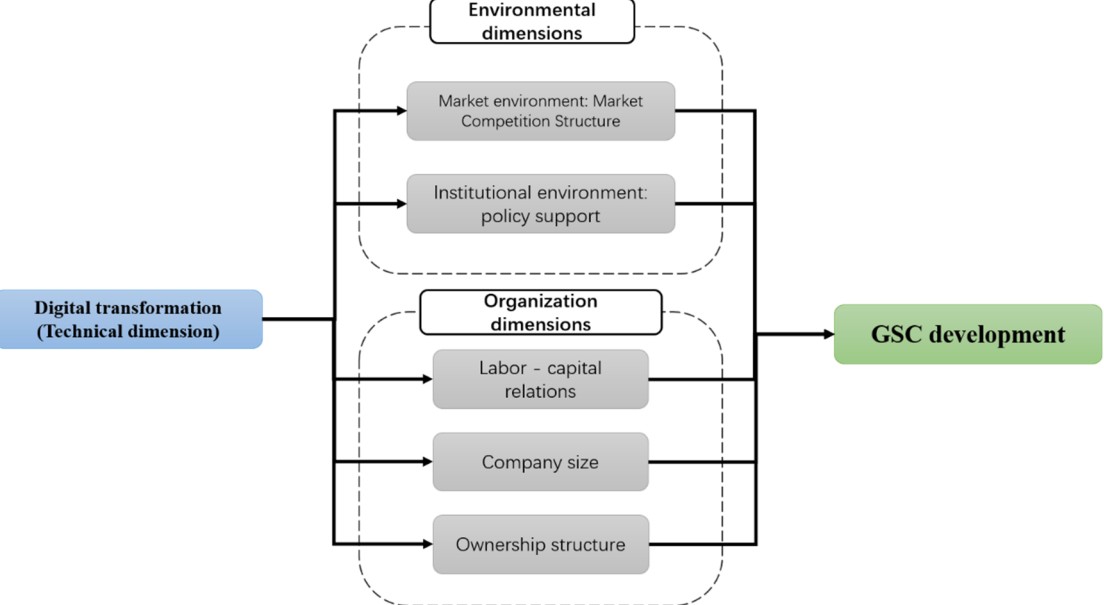

**Figure 1.** Digital transformation and green supply chain development in the TOE framework.

Firstly, digital technology has facilitated a more competitive market for sustainable products by enabling businesses to differentiate themselves based on their environmental performance. Consumers are becoming increasingly environmentally conscious, and businesses that can demonstrate their commitment to sustainability through their supply chains will have a competitive advantage in the market.

Secondly, governments around the world are implementing policies that incentivize businesses to adopt sustainable practices in their operations. For example, carbon taxes and emissions trading schemes encourage businesses to reduce their carbon footprint, while green procurement policies require businesses to source materials and products from environmentally friendly suppliers. Digital technology has enabled businesses to comply with these policies by providing them with tools to measure their environmental impact and track their progress towards meeting sustainability targets.

Thirdly, digital technology has also enabled businesses to organize their supply chains in a way that promotes sustainability. Through the use of supply chain management systems, businesses can monitor and optimize their operations in real-time, reducing waste and inefficiencies. Additionally, digital technology enables businesses to collaborate with their supply chain partners to share information on sustainability initiatives and best practices. This collaboration helps to align the goals of all parties involved, ensuring that sustainability is integrated throughout the entire supply chain.

Additionally, digital technology has enabled businesses to design and produce more sustainable products by providing them with tools such as lifecycle assessment software that evaluates the environmental impact of products throughout their entire lifecycle. This

information helps businesses make informed decisions about product design and material choices, resulting in more environmentally friendly products.

To summarize, digital technology has played a crucial role in driving green supply chain development through various channels including market competition, policy support, and business organization. The ability to differentiate oneself based on environmental performance in a competitive market, comply with sustainability policies, optimize supply chains, collaborate with partners, and design more sustainable products are all enabled by digital technology. As the world becomes increasingly focused on sustainability, the role of digital technology in green supply chain development is only set to grow in importance.

### 3.1. Environment Dimensions

### 3.1.1. Market Environment

The competitive market structures, such as oligopolies and monopolies, can have a significant impact on the contribution of digital transformation to green supply chains. This is because digital transformation requires significant investments that may not be feasible for smaller firms or those without significant market power. Additionally, larger firms with more market power may have less incentive to adopt sustainable practices in their supply chains, especially if they do not face significant competition.

In an oligopoly market structure, a small number of large firms dominate the industry. These firms often have significant market power, which means they can influence the price and quality of goods and services. When it comes to green supply chains, firms in an oligopoly may have less incentive to invest in digital transformation due to their market power. For example, in 2022, Air China, China Southern, and China Eastern will account for more than 80% of China's air passenger and cargo traffic. However, these large companies dominate the airline industry, and despite the environmental impact of air travel, there has been limited investment in sustainable aviation fuel technologies. This is partly because the cost of investments in these technologies would be spread across a smaller number of players in the industry, reducing the returns for any single firm.

In contrast, in a monopolistic market structure, one firm dominates the industry, with little competition. In this situation, the dominant player has the power to set prices and determine the terms of trade, making it difficult for new entrants to compete. Monopolies may have even less incentive to invest in digital transformation for green supply chains since they already hold significant market power. For example, in the logistics industry, some logistics giant companies have faced criticism for their carbon footprints, but their dominance in the market means that they face little pressure to adopt more sustainable practices.

However, it is worth noting that the role of market structure on digital transformation in green supply chains is not always straightforward. In some cases, competition can drive innovation and sustainability. For example, in the short-haul freight, electric trucks are becoming more prevalent, in part due to competition from new entrants like Geely and BYD in China. The threat of losing market shares to a smaller, more agile firm can drive established companies to invest in new technologies and sustainability initiatives.

Moreover, digital transformation can also create opportunities for smaller firms to compete with large ones by enabling them to access sustainability-related information and technology. For instance, small-scale farmers in China's rural areas can use mobile apps to access market information and sustainable farming practices, helping them to improve the agricultural supply chain while reducing their environmental impact. Based on the above discussion, we propose hypotheses about how the market environment affects the effect of digital transformation on green supply chains.

**H1.** *The higher the degree of market competition, the stronger the positive effect of digital transformation on green supply chain development.*

### 3.1.2. Institutional Environment

Policy factors play a crucial role in shaping the contribution of digital transformation to green supply chains in China. By creating incentives for businesses to adopt sustainable practices and providing support for investments in digital technologies, policies can accelerate the transition towards more sustainable supply chains. We will discuss some policy factors affecting the contribution of digital transformation to green supply chains in China, along with examples.

One important policy factor is government regulation. In China, the 13th Five-Year Plan (2016–2020) aimed to address environmental challenges by prioritizing environmental protection and ecological civilization. The plan included specific targets such as reducing energy consumption per unit of GDP by 15%, reducing carbon dioxide emissions per unit of GDP by 18%, and increasing forest coverage by 23%. These targets have been driving demand for digital transformation across industries, including supply chains. For instance, the "Green Fence" policy implemented by the Chinese government in 2013 has made it mandatory for all imported waste to meet high environmental standards, which has spurred investment in digital technologies like smart waste management and recycling systems.

Governments can also provide financial incentives for businesses to invest in digital transformation for green supply chains. In China, the central government has launched several initiatives to promote sustainable development, including the Green Credit Policy, which provides preferential loans to companies that adopt sustainable practices. Additionally, the National Development and Reform Commission (NDRC) has provided subsidies to encourage the adoption of renewable energy and energy-saving technology in industries, which has accelerated the uptake of digital technologies in supply chains.

Moreover, public procurement policies can drive demand for sustainable products and services and incentivize businesses to adopt sustainable practices. In China, the government has introduced policies to encourage the use of green products in public procurement. For example, the Ministry of Finance and the NDRC jointly issued the "Guidance on Promoting Green Procurement", which mandates that all government departments and institutions must purchase environmentally friendly products and services. This has incentivized businesses to adopt sustainable practices throughout their supply chain to meet the growing demand for green products and services.

Finally, industry-led initiatives can also play a role in promoting digital transformation in green supply chains in China. For instance, Alibaba's "Green Logistics" initiative aims to optimize logistics operations and reduce emissions by leveraging digital technologies such as big data analytics and IoT sensors. The company has invested in electric vehicles, smart warehouses, and renewable energy to support its commitment to sustainable logistics. Similarly, JD.com's "Green Stream Initiative" has been launched to build an eco-friendly supply chain ecosystem, which focuses on reducing waste and pollution while improving efficiency through digital transformation.

In conclusion, policy factors play an important role in driving the contribution of digital transformation to green supply chains in China. Government regulation, financial incentives, public procurement policies, and industry-led initiatives are all examples of policy tools that can accelerate the transition towards more sustainable supply chains. The Chinese government has implemented various policies and initiatives to promote sustainable development, which has incentivized businesses to invest in digital transformation to improve their environmental performance throughout the supply chain. Therefore, we propose a second hypothesis for this paper.

**H2.** *The higher the degree of policy support, the stronger the positive effect of digital transformation on green supply chain development.*

### 3.2. Organization Dimensions

In the organizational dimensions, labor–capital relations, company size, and ownership factors can all affect the contribution of digital transformation to green supply chains in China. Labor-intensive industries may face challenges in adopting digital technologies,

but worker training programs can help overcome these challenges. Smaller companies may have limited access to capital but can benefit from digital technologies that reduce costs and improve efficiency. State-owned enterprises may be subject to more regulatory pressure, but private enterprises may still invest in digital technologies to improve their environmental performance. Finally, ownership factors can impact the degree of collaboration and information sharing in supply chains, which can incentivize suppliers to adopt sustainable practices and invest in digital technologies.

Labor–capital relations play an important role in determining how digital technologies are adopted and used in supply chains. In China, labor-intensive industries such as textiles have faced challenges in adopting digital technologies due to high labor costs. However, labor–capital relations can be improved through worker training programs that promote the use of digital technology to enhance productivity and reduce environmental impacts. For example, the Green Textile Production program run by the International Finance Corporation (IFC) aims to promote the adoption of sustainable practices by training workers in energy-efficient production methods and the use of digital technologies like automation and data analytics. We, therefore, propose a theoretical hypothesis on the role of labor–capital relations in the process of digital transformation affecting the development of green supply chains (H3).

**H3.** *The positive effects of digital transformation on green supply chains are greater in labor-intensive firms than in capital-intensive firms.*

Company size is another factor that can impact the contribution of digital transformation to green supply chains in China. Smaller companies may face more challenges in adopting digital technologies due to limited resources and access to capital. However, digital technologies can help smaller companies overcome these limitations by reducing costs and improving efficiency. For instance, the Chinese startup Haier has developed a smart supply chain system that enables small and medium-sized enterprises (SMEs) to optimize their logistics operations and reduce waste through data analytics.

**H4.** *The positive effects of digital transformation on green supply chains are likely to be more evident in SMEs than in large enterprises.*

Ownership factors can also play a role in shaping the adoption of digital technologies in green supply chains. State-owned enterprises (SOEs) in China are subject to more stringent regulations and policies related to sustainability compared to private enterprises. This can create incentives for SOEs to invest in digital technologies that improve environmental outcomes. For example, the state-owned China National Chemical Corporation (ChemChina) has invested in digital technologies to monitor and reduce greenhouse gas emissions in its supply chain.

In contrast, private enterprises may face less pressure to adopt sustainable practices due to fewer regulatory requirements. However, private enterprises may still be motivated to invest in digital technologies to improve efficiency and reduce costs. For example, the Chinese e-commerce giant Alibaba has developed a green logistics network that leverages digital technologies like IoT sensors and big data analytics to optimize logistics operations and reduce emissions.

Moreover, ownership factors can also influence the degree of collaboration and information sharing in supply chains. In China, some multinational corporations have implemented sustainability initiatives that involve sharing best practices with suppliers to improve environmental outcomes. For instance, Walmart's Sustainability Consortium provides a platform for stakeholders to collaborate on sustainability issues and share information on best practices. This can incentivize suppliers to adopt sustainable practices and invest in digital technologies that enhance their environmental performance.

**H5.** *Ownership would affect the contribution of digital transformation to green supply chain development, but the direction of the impact is uncertain.*

## 4. Data and Measurements

In this section, we will explain the measures used in the empirical analysis of this paper and their data sources. In this section, we describe the measurements used in the empirical analysis of this paper and their data sources. This is followed by a description of the empirical research design in order to test the hypotheses presented in the theoretical analysis.

### 4.1. Manufacturing Digital Transformation

Considering the reality of production processes, different types of digital technologies are different for the development of green supply chains. As shown in Figure 2, the impact of these technologies may be divided into three areas: (1) technologies that can contribute to transport capacity; (2) technologies that can improve the efficiency of logistics management; (3) technologies that can reduce the carbon emissions of the transport process. Therefore, we divide digital transformation technologies into five categories and measure the level of digital transformation of these technologies separately.

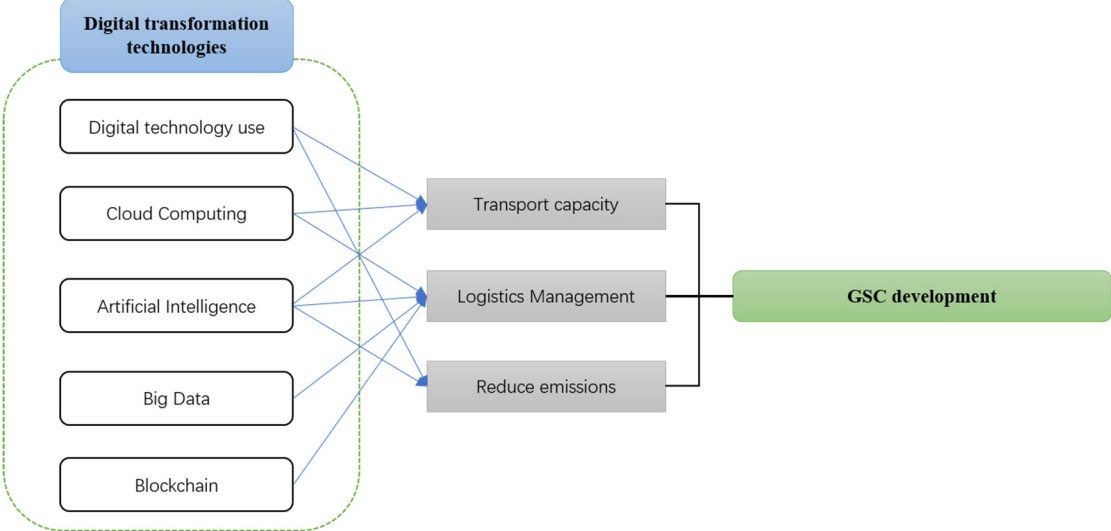

**Figure 2.** Types of digital technologies affecting the development of GSC.

Referring to the methods of existing studies [65–67], we used the Python crawler function and the Java PDFbox library to crawl the relevant vocabulary reflecting the digital transformation characteristics of enterprises from the annual reports of A-share listed manufacturing enterprises officially published by the stock exchanges in Shanghai and Shenzhen. Referring to the theoretical framework of Liu et al. [65] and Kong et al. [67], we divided the digital transformation index into five aspects: "big data technology, cloud computing, artificial intelligence, blockchain technology, and digital technology application", and calculated the number of keywords appearing in each aspect in this paper separately. Considering the development of digital transformation technology in recent years, we increased the existing lexicon from 79 terms to 156 terms. Finally, we used machine learning to perform semantic correction and synonym merging, and summed the number of keywords in the company's annual report to obtain the variables that reflect the level of digital transformation. For the brevity, we display the details of the lexicon in the appendix (see as Table A1). We summed these indices to the provincial level to obtain the digital transformation variables for the corresponding province, based on information on the domicile of A-share listed companies.

Because the impact of digital transformation and GSC development is closely related to the scale of the economy it reached, referring the related research [5,68], we calculated the weighted average of the digital transformation index with provincial-level GDP as the weight (Equation (1)). GDP data from China Statistical Yearbook. Our independent

variables used the aggregate index and the sub-index of different types of digital technology.

$$DT_{i,j,t} = \frac{\left(GDP_{i,t} \times digital\ transformation_{i,t} + GDP_{j,t} \times digital\ transformation_{j,t}\right)}{GDP_{i,t} + GDP_{j,t}}$$

### 4.2. Regional Market and Policy Environment Characteristics

According to the theoretical model, we need to measure the competitive environment of the market. According to Grančay et.al. [69] and He et al. [70], market potential is inversely related to the degree of market competition (MC). Therefore, the inverse of the market potential can be used as an indicator of the degree of market competition. A larger MC indicator means that the market structure of the region is more monopolistic, while conversely the market environment is more competitive. The variable measure reflecting market competition is shown in Equation (2):

$$MC_{i,t} = \frac{1}{\sum_{j \neq i}\left(\frac{Y_{jt}}{d_{jt}}\right) + Y_{it}/d_{ii}} * 100 \tag{1}$$

The competitive market environment index is calculated from GDP and geographical distance, which represents the size of a region's potential domestic market. In Equation (1), $Y_{it}$ represents the real GDP of $i$ province in year $t$. $d_{ij}$ represents the transportation distance between two provinces $i$ and province $j$, which is measured by the map distance between two provincial capitals. About the market potential intern province ($d_{ii}$), the calculation is from He et al. [70] and Li et al. [5] research, as is the following Equation (3), where, $area_i$ represents the area of province $i$ and the circumference is taken as $\pi = 3.14$. Map distances are taken from Google Maps.

$$d_{ii} = \frac{2}{3} * \sqrt{area_i/\pi} \tag{2}$$

At the same time, according to the theoretical framework in Section 3, we need a variable to measure the institutional environment regarding GSC development. The intensity of environmental regulation is an important institutional context that influences the development of GSC. Measuring the level of support for environmental regulation policies can be a complex task that involves a range of different indicators and methods. Some possible approaches include public opinion surveys, legislative analysis, regulatory compliance, and so on. However, it is important to note that measuring support for environmental regulation policies can be subjective and influenced by a variety of factors, such as political ideology, cultural values, and personal beliefs. Therefore, in order to guarantee the objectivity of the measurement, we use the relative size of the more environmental governance inputs to measure the regional environmental regulation strength. The scale of investment in environmental governance may not be reflective of the intensity of environmental regulation, given the vast differences in the size of China's provincial economies. Therefore, we choose the total investment in industrial pollution control as a proportion of industrial value added as a measure of environmental regulation (ER). The above data are taken from the China Statistical Yearbook and the China Industrial Statistics Yearbook.

$$ER_{i,t} = \frac{Investment\ completed\ in\ industrial\ pollution\ control_{i,t}}{industrial\ value\ added_{i,t}} * 100 \tag{3}$$

### 4.3. Measurements on Organizational Dimension Variables

We chose three indicators to measure the characteristics of the organizational dimension separately as mentioned in Section 3.2. Firstly, we use the wage-to-asset ratio of scale industrial enterprises of the corresponding region to reflect the labor–capital relations (LCR). The labor–capital ratio is widely used in economic research to reflect the relationship between workers and owners within a firm [71,72]. According to the standards of the China Bureau of Statistics, scale industrial enterprises in China are those with an annual



main business revenue of 20 million yuan or more. These raw data come from industrial yearbooks and the official website of each provincial statistical office, and the labor–capital ratios are calculated by the authors. Secondly, referring to existing studies on Chinese enterprises [73–76], we introduce the amount of fixed asset investment in scale industrial enterprises of the corresponding region to reflect the size of the enterprise (SE). The raw data are obtained from the China Statistical Yearbook and the China Industrial Statistical Yearbook. Third, we choose the share of state-owned enterprises in scale industrial enterprises to reflect the ownership characteristics of the region (OS). The raw data are obtained from the China State-owned Assets Supervision and Administration Yearbook and the indicator is calculated by the author.

### 4.4. Green Supply Chain Development

Studies have been conducted to measure the level of development of the green supply chain in terms of carbon emissions from the transportation sector [1,77]. However, GSC Green supply chain development needs to take into account both environmental and economic factors [78]. Therefore, we calculate $CO_2$ emissions per unit of trade as an index reflecting the level of GSC development (GSC) as Equation (5). In order to make this variable more intuitive to reflect the transportation carbon reduction, we use its opposite in the regression.

$$\text{GSC}_{i,j,t} = -\frac{carbon\ emission_{i,t} + carbon\ emission_{j,t}}{transport\ volume_{i,j,t}} \tag{4}$$

In order to improve the comparability of this paper, we choose the bilateral inter-provincial railway traffic volume as a proxy variable reflecting the transport volume (*transport volume*, unit: ton). China's railway carriers are mainly state-owned enterprises, and the data on railway traffic volume are more accurate by China's official statistics; therefore, existing studies have chosen railway traffic volume as a proxy variable for traffic volume [5,76]. The data are from China Railway Yearbook. Meanwhile, we choose the $CO_2$ emissions from the transport sector as a proxy for carbon emissions (*carbon emission*, unit: kg), and these data come from the China Energy Yearbook. In Equation (5), *i,j,t* denote the departure province (*i*), destination province (*j*) and year (*t*), respectively. Therefore, the carbon emission status of railway transportation is selected to reflect the level of green supply chain development in China. We collected data from 2002 to 2017 and the statistical characteristics of all the variables used in the study of this paper are shown in Table 1.

**Table 1.** Statistical characteristics of variables.

| Variable | Definition | Mean | Std. Dev. |
|---|---|---|---|
| GCD | Level of GSC development | −0.26 | −0.10 |
| DT | Level of digital transformation | 9767.54 | 1083.24 |
| MC | Proxy of market competition defined in Section 4.2 | 0.03 | 0.0006 |
| ER | Proxy of environment regulation defined in Section 4.2 | 0.04 | 0.0025 |
| LCR | The wage-to-asset ratio of scale industrial enterprises | 0.20 | 0.11 |
| SE | fixed asset investment in scale industrial enterprises (unit: 100 million CNY) | 10,054.37 | 2918.84 |
| OS | The share of state-owned enterprises in scale industrial enterprise | 0.58 | 0.15 |
| dis | The geography distance between provincial capital in the Google map (unit: km) | 1305.93 | 744.09 |

## 5. Empirical Studies

### 5.1. Methodology

#### 5.1.1. Test method on Environmental Dimension

Referring to Anderson et al. [79], Fang et al. [80], and Li et al. (2022) [5], we use the following regression to examine the influence of market environment and policy environment on the development of green supply chain (H1 and H2). In the Equation (5), $lnGCD_{i,j,t}$, $lnDT_{i,j,t}$ and $lndis_{i,j}$ are the logarithmic forms of the variables described in the previous

sub-section; $\mu_i$, $\mu_j$ are the fixed effects of region $i$ and region $j$, respectively; $\alpha$, $\varepsilon_{i,j,t}$ are the intercepts of the error term. $\beta_1$ and $\beta_2$ describes parameters that need to be estimated.

$$lnGCD_{i,j,t} = \alpha + \beta_1 lnDT_{i,j,t} + \beta_2 lndis_{i,j} + \mu_i + \mu_j + \varepsilon_{i,j,t} \tag{5}$$

Further, we examine the heterogeneity in the moderating effects of market competition and policy support. We divide regions into three groups according to three percentiles of the MC and ER defined above, lower than 25% (low group), 25% to 75% (mid group), and higher than 75% (high group). We will analyze the moderating effect of market competition and policy environment by conducting regressions as in Equation (6) under each of the three types of subsamples and comparing the regression results. If the coefficient $\beta_1$ is higher in the high group than in the other groups, then hypotheses H1 and H2 can be proved, and conversely, hypotheses H1 and H2 may not be valid.

5.1.2. Test Method on Environmental Dimension

We test the hypotheses about the organizational dimensions by introducing interaction terms (H3, H4, and H5). The regression is shown in Equation (7).

$$\begin{aligned} lnGSC_{i,j,t} = \alpha + \beta_1 lnDT_{i,j,t} + \beta_2 lnMechanism\ variables_{i,j,t} + \beta_3 lnDT_{i,j,t} \\ * lnMechanism\ variables_{i,j,t} + \beta_4 lndis_{i,j} + \mu_i + \mu_j + \varepsilon_{i,j,t} \end{aligned} \tag{6}$$

In Equation (7), we introduce mechanism variables that respond to labor relations, firm size, and ownership, and include interaction terms between these mechanism variables and the digital transformation independent variables. The rest of the settings are consistent with Equation (6).

*5.2. Empirical Results on the Environmental Dimension*

We test hypotheses about the environment dimensions according to the methodology in Section 5.1.1. We separately estimate baseline results based on the subsamples and plot the estimated coefficients and 95% confidence intervals in Figure 3. Figure 3a shows the regression coefficients on digital transformation and GSC development for different sub-samples with different degrees of market competition; Figure 3b shows the regression coefficients on digital transformation and GSC development for different sub-samples with different policy support (environmental regulation). In Figure 3, the bule bars show the results for the prefectures with centrality lower than the 25th percentile, with the red bars for those between the 25th and 75th percentiles, and green bars for those higher than the 75th percentile.

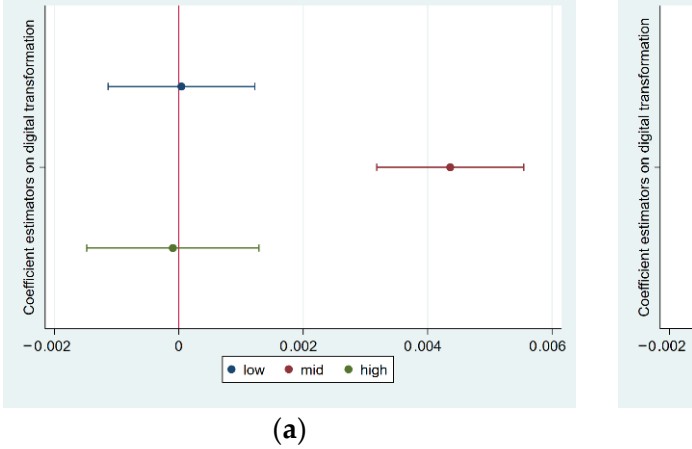
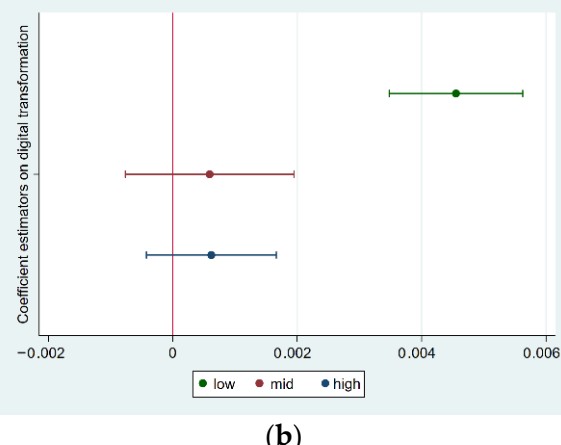

(**a**)                                        (**b**)

**Figure 3.** Regression coefficients and 95% confidence intervals in different subsamples of market competition (**a**) and environmental regulation (**b**).

First, based on the results in Figure 3a, we find that the coefficient on digital transformation is significantly non-zero only in the group with a moderate level of market competition ($\beta_1 = 0.0042$), while it is not significant in both the groups with lower and higher levels of market competition. It is not the case that the effect of digital transformation on green supply chain development is stronger in regions with higher levels of market competition, which does not support Hypothesis H1.

This result suggests that only a moderately competitive market environment can leverage the promoting effect of digital transformation on GSC development. On the one hand, when the market is in high monopoly status, monopoly companies have little incentive to digitally transform or take decisions that promote the development of green supply chains. This makes the promoting effect of digital transformation to GSC development insignificant at lower levels of market competition. On the other hand, when the market is highly competitive, firms usually want to adopt business strategies that can control costs in the short term, rather than undertake costly R&D activities on digitalization or GSC development.

Second, according to Figure 3b, we find that the group with the highest level of policy support (with the strongest environmental regulation) has the most significant contribution of digital transformation to GSC development ($\beta_1 = 0.0045$). This result means that the higher the level of policy support regarding environmental protection for a region, the more pronounced the promoting effect of digital transformation on GSC development, which supports the H2 in Section 3.1.2.

Third, in order to analyze the different impacts of different digitization technology types, we replace the independent variable (DT) with sub-indexes as described in the Section 4.1. Figure 4 displays the performance of the sub-indexes in different market competition levels, where Figure 4a,b show the coefficients corresponding to each type of sub-index with 95% confidence intervals in the low market competition level and medium market competition level, respectively.

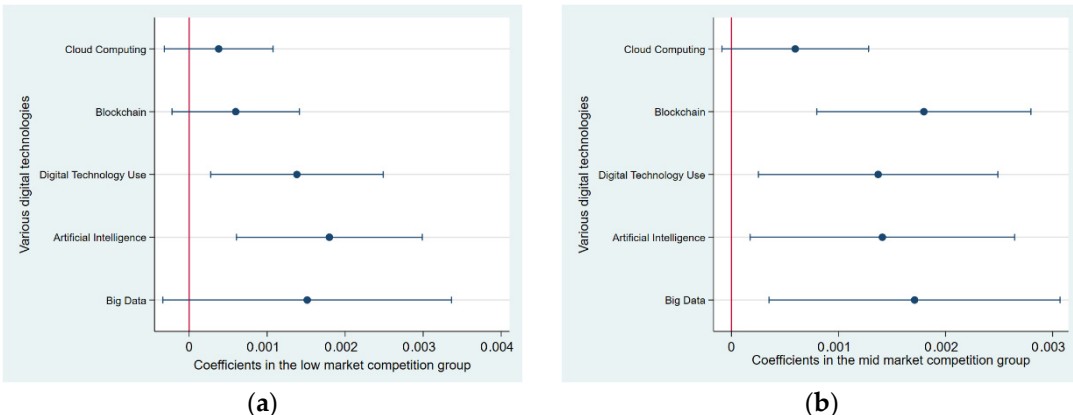

**Figure 4.** Regression coefficients and 95% confidence intervals for various sub-indexes of digital technologies in different subsamples of market competition ((**a**) is in low group and (**b**) is high).

It is worth noting that regardless of the degree of market competition in the environment, the promotion of cloud computing for the development of GSC is not significant in the current China. We suggest that there are several possible reasons: Firstly, cloud computing technology requires large-scale data centers to support data storage and processing needs. These data centers often require significant energy supply and operational maintenance, including electricity, air conditioning, and network equipment. If these data centers rely on non-renewable energy sources, such as coal or oil, they may have a negative environmental impact, leading to increased carbon emissions, which hinders the promotion of a truly green supply chain. As of 2022, China's largest source of electricity continues to be coal power, which accounts for 58.4% of total power supply. As a result, large computing cloud technology centers still rely on coal power, a category that con-

tributes to significant carbon emissions. Secondly, cloud computing technology involves data transmission and network latency issues. Large-scale data transmission consumes substantial amounts of energy, particularly when it involves transferring data across different regions or countries. Furthermore, as cloud computing involves data transmission over networks, network latency can result in reduced efficiency and energy wastage, further impeding the progress of green supply chains. Thirdly, cloud computing technology needs to adhere to local regulations and compliance requirements. China has strict requirements and policies regarding green environmental practices, including energy consumption, e-waste management, and data privacy. Cloud service providers must comply with these regulations and ensure data security and privacy, which can increase the complexity and cost of technical implementation.

Another interesting phenomenon is that the role of blockchain and big data technologies is not significant in regions with low levels of market competition, while it is significant in regions with moderate levels of market competition. The possible reason for this phenomenon is that there are differences in the needs for GSC development under different market competition structures. Blockchain technology ensures traceability, transparency, and accountability in the green supply chain, while big data technology enables data analytics, risk mitigation, and collaboration. Together, they empower stakeholders to make sustainable decisions, drive efficiency, and foster a greener and more responsible supply chain. These applications tend to have the effect of promoting competition in the market, and, therefore, in highly monopolized market structures, monopolies do not want to develop such competition-oriented technologies.

Similarly, Figure 5 displays the performance of the sub-indexes in different market competition levels, where Figure 5a,b show the coefficients corresponding to each type of sub-index with 95% confidence intervals in the low policy support level and high policy support level, respectively.

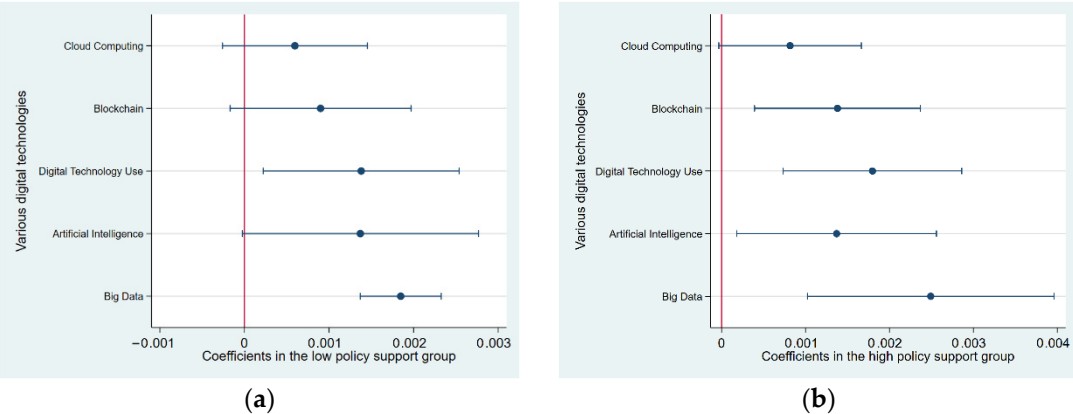

(**a**)          (**b**)

**Figure 5.** Regression coefficients and 95% confidence intervals for various sub-indexes of digital technologies in different subsamples of policy support. ((**a**) is low group and (**b**) is mid).

By comparing Figures 4 and 5, we find that the application side of digital technology (digital technology use, artificial intelligence) is the main type of digital technology that promotes the development of GSC in China at present, both under different competitive market environments and different levels of policy support. The role of more upstream digital technologies, such as cloud computing, in promoting the development of GSC technology is not significant.

### 5.3. Empirical Results on the Organizational Dimension

We test the effect of the organizational dimension according to the methodology in Section 5.1.2. The regression results are shown in Table 2, and for robustness reasons, we report the regression results with and without the interaction term, respectively. Based on the regression results in Table 2, we get the following three conclusions.

**Table 2.** Regression Results Including Organizational Dimension Variables.

| | Dependent Variable: log(GSC) | | | | | |
|---|---|---|---|---|---|---|
| **Equations** | **(1)** | **(2)** | **(3)** | **(4)** | **(5)** | **(6)** |
| DT | 0.003 ** | 0.002 ** | 0.003 *** | 0.003 *** | 0.003 *** | 0.002 *** |
| | (1.98) | (2.02) | (2.39) | (2.35) | (3.05) | (2.98) |
| LCR | −0.219 *** | −0.189 *** | | | | |
| | (−4.27) | (−3.60) | | | | |
| DT × LCR | | 0.025 ** | | | | |
| | | (2.00) | | | | |
| SE | | | 0.188 *** | 0.181 *** | | |
| | | | (2.99) | (3.05) | | |
| DT × SE | | | | 0.004 | | |
| | | | | (1.50) | | |
| OS | | | | | 0.099 *** | 0.085 *** |
| | | | | | (2.93) | (2.38) |
| DT × OS | | | | | | 0.018 *** |
| | | | | | | (2.77) |
| lndis | −0.699 ** | −0.699 * | −0.717 * | −0.720 | −0.733 ** | −0.709 * |
| | (−1.97) | (−1.88) | (−1.85) | (−1.42) | (−1.97) | (−1.88) |
| intercept | 6.546 *** | 5.543 *** | 12.338 *** | 9.472 *** | 6.646 *** | 6.620 *** |
| | (11.76) | (16.70) | (11.45) | (18.27) | (11.38) | (13.16) |
| Region *i* FE | Yes | Yes | Yes | Yes | Yes | Yes |
| Region *j* FE | Yes | Yes | Yes | Yes | Yes | Yes |
| F | 23.55 *** | 23.53 *** | 32.80 *** | 36.48 *** | 22.40 *** | 25.95 *** |
| Ajusted $R^2$ | 0.54 | 0.56 | 0.65 | 0.65 | 0.47 | 0.44 |

Note: The t-values corresponding to the coefficients are in parentheses and ***, **, * represent the 1%, 5%, and 10% levels of significance, respectively.

First, we find that the coefficient of the variable LCR is significantly negative in Equations (1) and (2), which indicates that supply chain carbon emissions will be higher in regions with more labor-intensive firms relative to regions with more capital-intensive firms. Meanwhile, the interaction term between LCR and DT is positive, indicating that digital technology contributes more significantly to GSC development in regions with more labor-intensive firms, which is consistent with H3. As mentioned in the theoretical analysis in Section 3.2, labor-intensive firms, which rely heavily on human resources, often have more dynamic and flexible operations compared to capital-intensive firms. Digital technologies, such as automation, robotics, and AI-powered systems, can be easily integrated into their operations, enabling efficient resource utilization and process optimization. This flexibility allows labor-intensive firms to adapt and implement sustainable practices more swiftly, resulting in a more effective promotion of green supply chain development. Meanwhile, digital technologies often require substantial upfront investment and technological infrastructure. Capital-intensive firms, which heavily rely on expensive machinery and equipment, may face challenges in adopting and integrating new digital technologies due to the complexity and cost involved. Additionally, the lifecycle of capital-intensive assets tends to be longer, making it more difficult to replace or upgrade them with newer, more sustainable technologies. This can hinder the effective implementation of digital technologies for green supply chain development in capital-intensive firms. It is important to note that while digital technologies may have more immediate and direct impact in labor-intensive firms, capital-intensive firms can still leverage digital solutions for sustainability improvements. This could include implementing advanced data analytics to optimize resource allocation, adopting IoT-enabled systems for energy monitoring, or utilizing blockchain technology for transparency and traceability.

Second, the empirical results do not support the content of H4. In Equation (4), the interaction term between DT and SE is not significant, which indicates that the firm size does not have a significant effect on the effectiveness of digital technology for GSC development. However, this finding does not mean that firm size has no effect on GSC development. On the contrary, according to the results of Equation (3), 1% increase in the average regional manufacturing firm size could reduce supply chain carbon emissions by approximately 0.188%. The above results suggest that the scaling up of firms has a positive effect on GSC

development, but this positive effect is not realized through digital transformation. This conclusion can be corroborated with the conclusion on market competition in the previous subsection of this paper. In China, larger manufacturing firms tend to be monopolistic in nature, and these firms have no incentive to develop digital transformation technologies due to their monopolistic market position.

Third, regarding Hypothesis 5, the empirical results in Equations (5) and (6) show that SOEs are better able to utilize digital technology to promote the GCS development. Based on the theoretical analysis in Section 3, there may be several practical reasons for this result: (1) Government support and political resources: In China, SOEs often operate with the backing and support of the government. They can benefit from government policies, incentives, and funding dedicated to promoting sustainable development and green initiatives. This support provides SOEs with access to resources and expertise that can facilitate the adoption and integration of digital technologies for green supply chains. It also enables collaboration between SOEs and government agencies, fostering a conducive environment for sustainable practices. (2) Long-term vision: SOEs often have a long-term vision aligned with government objectives. This long-term perspective allows them to invest in and commit to sustainable initiatives, including the adoption of digital technologies. Unlike some private enterprises that may prioritize short-term profits, SOEs can focus on long-term environmental and social sustainability goals. This stability and continuity can drive consistent efforts towards building green supply chains by effectively leveraging digital technologies. (3) Regulatory compliance: SOEs are often subject to specific regulatory requirements and obligations. Governments may enforce stricter environmental regulations and standards on these enterprises, which can include mandates for green supply chain practices. This regulatory framework creates a conducive environment for SOEs to adopt and implement digital technologies that facilitate compliance and promote sustainability. Compliance-driven initiatives can further enhance the development of green supply chains within SOEs.

## 6. Conclusions

This paper focuses on how the companies' organization structure and the socio-economic environment interact with digital technologies under the process of green supply chain development. Based on the "Technology-Organization-Environment" (TOE) framework, this paper analyzes how digital transformation can drive green supply chain development. To test the TOE theoretical analysis framework, this paper calculates the digital transformation and green supply chain development index at the provincial level in China and conducts an empirical study. All theoretical hypotheses with their corresponding empirical results are summarized in Figure 6. In Figure 6, "+" indicates that the empirical study found that the factor has a positive impact on the interaction between digital transformation and green supply chain development, and "UN" indicates that the empirical results of the factor are uncertain.

The main findings and implications of this paper can be summarized in the following aspects:

First, according to the TOE theory, the external environment dimensions, such as the market and policy environments, affect the role of digital technology in promoting GSC development. On one hand, empirical results suggest that there is a non-linear relationship between the level of market competition and the role of digital technology on the GSC development. When the market is in high monopoly status, monopoly companies have little incentive to digitally transform or take decisions that promote the development of green supply chains. This makes the promoting effect of digital transformation to GSC development insignificant at lower levels of market competition. However, when the market is highly competitive, firms usually want to adopt business strategies that can control costs in the short term, rather than undertake costly R&D activities on digitalization or GSC development. Only a moderately competitive market environment can leverage the promoting effect of digital transformation on the GSC development. On the other

hand, in terms of policy environment, the higher the level of policy support regarding environmental protection for a region, the more pronounced the promoting effect of digital transformation on GSC development.

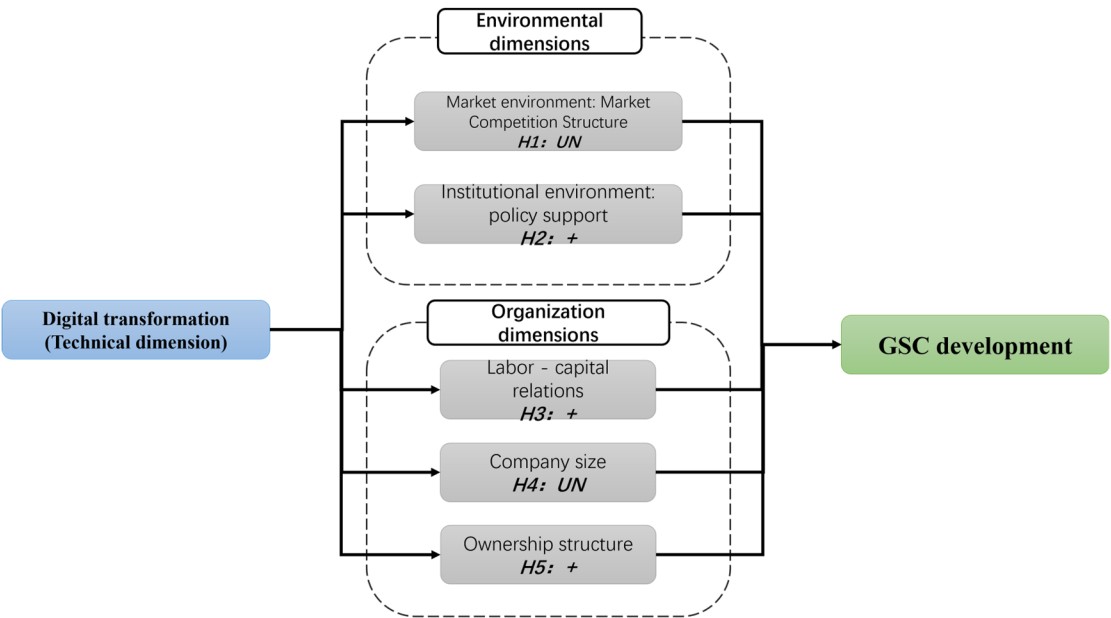

**Figure 6.** Summary of empirical findings.

Second, in the organizational dimensions, labor–capital relations, company size, and ownership factors can all affect the contribution of digital transformation to green supply chains in China. As for labor–capital relations, we find that supply chain carbon emissions will be higher in regions with more labor-intensive firms relative to regions with more capital-intensive firms. Regarding the ownership, the empirical results suggest that SOEs are better able to utilize digital technology to promote the GCS development. However, this paper finds that the firm size does not have a significant effect on the effectiveness of digital technology for GSC development.

Third, there are differences in the impact of different types of digitization technologies on GSC development. The application side of digital technology (digital technology use, artificial intelligence) is the main type of digital technology that promotes the development of GSC in China at present, both under different competitive market environments and different levels of policy support. The role of more upstream digital technologies, such as cloud computing, in promoting the development of GSC technology is not significant. The implication of this research finding is that the current application of digital technology in China should be more balanced in the area of green supply chain development. In particular, it should focus on the development of upstream technologies such as cloud computing, big data, and other fundamental technologies.

The study of this paper could provide some policy implications into achieving emission reductions in the transport sector through digital technology. For example, according to findings on external environment dimensions, this paper suggest that governments can foster an environment where digital technologies drive green supply chain development while maintaining healthy market competition by implementing policies. It encourages innovation, collaboration, and sustainable practices, leading to a more environmentally responsible and efficient supply chain ecosystem. What is more, findings related to organizational dimensions reveal that policymakers need to tailor their policies to promote the digital transformation based on the characteristics of different firms, such as capital structure, ownership structure, and other characteristics, in order to drive GSC development. Additionally, this paper's research on China's digital transformation and green supply

chains can also provide some insights for other countries that have the pressure to reduce transport emissions.

Although this paper provides a detailed analysis of the impact of digital transformation on green supply chain development and its mechanism, there are two main limitations of our study that can be improved in future studies. On one hand, due to lack of data, the green supply development data only include railway transportation. While rail transportation tends to be associated with the transportation of bulk commodities, road transportation tends to be more relevant to supply chains for retail, consumer goods, and small-scale intermediate goods as has already been mentioned in the research [35,80,81]. On the other hand, the data on digital transformation of this paper contain only the listed company sample, and it is also an interesting topic whether these effects exist in other unlisted firms, e.g., family firms, SMEs [9], etc.

**Author Contributions:** Conceptualization, W.L. and L.L.; methodology, X.X. and X.Y.; formal analysis, L.L. and X.Y.; data curation, X.X. and L.L.; writing—original draft preparation, W.L. and X.Y.; writing—review and editing, W.L., X.X., X.Y. and L.L.; supervision, W.L.; project administration, W.L. All authors have read and agreed to the published version of the manuscript.

**Funding:** This research received no external funding.

**Data Availability Statement:** The data are contained within the article.

**Conflicts of Interest:** The authors declare no conflict of interest.

## Appendix A

**Table A1.** Details of the digital transformation depth lexicon used in this paper.

| Digital Transformation Area | Core Vocabulary Used in Text Analysis | The Function of Technology in the Enterprise (Example) |
| --- | --- | --- |
| Digital technology use | Words describing specific applications or specific scenarios of Internet technology, such as e-commerce, industrial Internet, digital marketing, smart factory, etc. (77 words in total) | Drive business model innovation |
| Cloud Computing Technology | Words describing cloud computing technologies and applications such as cloud computing, cloud storage, industrial cloud, graph computing, physical information systems, etc. (18 words in total) | Reduce corporate sales costs |
| Artificial Intelligence Technology | Vocabulary that describes artificial intelligence technologies and applications such as business intelligence, semantic recognition, deep learning, etc. (24 words in total) | Improve data analysis capabilities; Reduce administrative costs |
| Big Data Technology | Vocabulary describing Big Data technologies and applications such as Big Data, digital platforms, integrated systems, information terminals, etc. (32 words in total) | Precision marketing; Optimization of management processes |
| Blockchain Technology | Vocabulary describing Block chain technologies and applications such as blockchain, digital currency, distributed computing etc. (5 words in total) | Anti-counterfeiting |

Note: For the sake of simplicity, all terms are listed. If you are interested in this thesaurus, please feel free to contact the author.

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
