# Peer review of "How Does Digital Transformation Impact Green Supply Chain Development? An Empirical Analysis Based on the TOE Theoretical Framework"

_systems, doi:10.3390/systems11080416_

Round 1

Reviewer 1 Report

 This paper focus on how the companies’ organization structure and the socio-economic environment interact with digital technologies under the process of green supply chain development. This study is interesting and meaningful.

Some issues are concerned.

1. The research object is the development of green supply chain, and the research method is the structural equation method. Please strengthen in the introduction that this article belongs to the category of Systems journal;

2. The collection of data is very important in this article, but the description of how to collect the data and how to structure the data for use in this study is unclear;

3. All formulas should be followed by Punctuation;

4. When labeling literature, only one system requirement needs to be met, such as "Liu et al. (2023) [58] and Kong et al. (2023) [60]" which should be "Liu et al. [58] and Kong et al. [60]";

5. After data analysis, some specific policies or suggestions should be put forward to help enterprises promote green supply chain development through Digital transformation;

6. Papers should appropriately reduce self-evaluation, such as ”The study of this paper can provide some policy implications into achieving emissions reductions in the transport sector through digital technology. Additionally, this paper's research on China's digital transformation and green supply chains can also provide some insights for other countries that have the pressure to reduce transport emissions“ in the abstract.

Reviewer 2 Report

The paper draws attention to a very interesting research field and subject. Its relevance to the role of digital transformation in green supply chain development makes it even more interesting. Yet, the paper requires major some changes as follows:

In the abstract session, I would suggest to better highlight the aim of this research together with its significance.

Research gap: In general, a good discussion is noted to present the need for this study. However, there is room to further underpin and support your arguments. Supporting the core arguments related to the research gap based on more recent literature is instrumental to convey the message further related to the core value of your study.

Considering the nascent field of “digital transformation and supply chain management”, the range of extant literature covered is apt and wide. I would welcome just a few more contemporary works, such as:

Empowerment and performance in SMEs: Examining the effect of employees' ethical values and emotional intelligence”, chapter in book: A Guide to Planning and Managing Open Innovative Ecosystems, Emerald Publishing Limited, ISBN 9781789734102

Corporate Environmental Responsibility, Accounting and Corporate Finance in the EU: A Quantitative Analysis Approach”, Springer International Publishing

The chosen mixed approach, instrument, and data analysis need further analysis.

The section of results was quite clear and well written and presented.

Discussion and Implications: There was a bit of a reach here in terms of providing conclusions that were not a direct result of the analyses. Some conclusions were also made indicating unique results related to field. These distinctions were not supported or justified by the study. I think the authors need to make sure the conclusions that are presented are based on the individual results from each hypothesis and that overreaches are not made

I thank the author(s) for the opportunity to read this interesting article and I hope that the above recommendations shall assist them in improving it.

An extended proofreading by a native English speaker is recommended. Some parts need a more "formal" approach.

Reviewer 3 Report

The authors present a very interesting research to understand how the digital transformation impact green supply chain development based on the TOE theoretical 3 framework. The work is well structured and very clear. Nevertheless some improvements should be made to the document:

1-     In the abstract is missing the reason for the research conducted. Where is the problem statement? Where is the underlining reason for the conducted research? The authors should be more clear about this topic to provide the reader with a conducting line that connects all the different stages of the research.

2-     The authors have two times the 5.2 -  5.2. Empirical results on the environmental dimension, and 5.2. Empirical results on the organizational dimension. Please correct this issue.

3-     The authors should provide a table to summarize the main 3 findings mentioned in the abstract. These could be illustrated in the conclusions section.

4-     The authors should develop the first conclusion in  a more detailed way: First, according to the TOE theory, the external environment dimensions, such as the market and policy environments, affects the role of digital technology in promoting GSC development.

5-     The conclusions should be divided into managerial implications, academic implications, and future research.

Good luck

Reviewer 4 Report

Dear authors,

I appreciated the content of the paper. However, a few changes are necessary:

·       Line 151: the title should be reviewed

·       Within text references should be written in the same way in the paper.

·       Considering hypothesis 5, it is not appropriate to formulate it as it's difficult to be determined theoretically. If that theoretical arguments are not enough to propose a certain direction, it would be better the hypothesis states just that the ownership may determine differences in the influence of digital transformation on green supply chain.

Reviewer 5 Report

Dear Authors

Indeed, digital technologies are having some positive impact on sustainability goals. In the transport industry, digital optimisation is important in green supply chain development processes. However, it is good that the authors also point out that digital, data processing technologies require large data centres, which in turn require the provision of large amounts of energy. The authors point out that in China, more than 50% of energy still comes from non-renewable sources, mainly coal, which generates climate and environmental problems. It is important that the authors note this fact. Nevertheless, the huge amounts of greenhouse gas emissions emitted into the atmosphere and the climate crisis call for research that can contribute to their reduction. The article presented for review is such an article. Thus, the topic addressed is important and timely. 

The article is also well structured. 

I have a few comments that the Authors should include in the revisions:

1). it needs to be explained in detail why such and not other indicators were chosen for the analyses. Here laconic one-sentence justifications appear. This should be expanded, as the choice of indicators sometimes fundamentally affects the results of the research. The objectivity of the research undertaken needs to be justified. 

2) It might be worth adding a comparative study of car and rail transport, or at least mentioning it ( https://doi.org/10.3390/en14216875 ). This is a very important aspect affecting climate and environmental protection. 

3) In the literature review section, the authors present the broad spectrum of digital applications. It is worth expanding the literature here to include researchers from other geographical regions. Studies of this type are plentiful in the literature. It is worth including, for example: 10.3390/en12173289 ; 10.34021/ve.2022.05.03(4) ; 10.3390/en12203891 ; 10.30657/pea.2022.28.50   And many others. 

4) The contribution of individual authors to the preparation of this manuscript is not specified. To be completed. 

Best of luk

Reviewer

  •  

Round 2

Reviewer 1 Report

This paper can be accepted now.

Reviewer 3 Report

Good job!